# Cardiotoxicity Induced by Immune Checkpoint Inhibitors: What a Cardio-Oncology Team Should Know and Do

**DOI:** 10.3390/cancers14215403

**Published:** 2022-11-02

**Authors:** Concetta Zito, Roberta Manganaro, Giuliana Ciappina, Calogera Claudia Spagnolo, Vito Racanelli, Mariacarmela Santarpia, Nicola Silvestris, Scipione Carerj

**Affiliations:** 1Cardiology Unit, Department of Clinical and Experimental Medicine, University of Messina, 98125 Messina, Italy; 2Medical Oncology Unit, Department of Human Pathology “G. Barresi”, University of Messina, 98125 Messina, Italy; 3Department of Interdisciplinary Medicine, Medical School, University of Bari “Aldo Moro”, 70121 Bari, Italy

**Keywords:** immunotherapy, immune checkpoint inhibitors (ICIs), predictive biomarkers, multidisciplinary treatments, immune-related toxicity, cardiotoxicity, cardioncology

## Abstract

**Simple Summary:**

Immune checkpoint inhibitors, including monoclonal antibodies directed against the programmed cell death-1 (PD-1)/programmed cell death ligand-1 (PD-L1) pathway or the cytotoxic T-lymphocyte antigen-4 (CTLA-4) pathway and the most recent lymphocyte-activation gene 3 (LAG-3)—blocking antibody, currently represent the standard of care for the treatment of a large number of solid tumors. However, the development of immune-related adverse events can limit the use of these beneficial agents in the clinical setting. It is of crucial importance to identify predictive biomarkers for those patients most likely benefiting from immunotherapy.

**Abstract:**

Immune checkpoint inhibitors (ICIs) have revolutionized the therapeutic scenario for several malignancies. However, they can be responsible for immune-related adverse events (irAEs), involving several organs, with a pooled incidence ranging between 54% and 76%. The frequency of cardiovascular system involvement is <1%. Among the cardiovascular irAEs, myocarditis is the most common and the most dangerous but other, less common manifestations of ICI-related cardiotoxicity include pericardial disease, arrhythmias, Takotsubo-like syndrome, and acute myocardial infarction, all of which remain poorly explored. Both oncologists and cardiologists, as well as the patients, should be aware of the possible occurrence of one or more of these complications, which in some cases are fatal, in order to implement effective strategies of cardiac surveillance. In this review, we summarize the latest studies and recommendations on the pathogenesis, clinical manifestation, diagnosis, and management of ICI-related cardiotoxicity in order to realize a complete and updated overview on the main aspects of ICI-related cardiotoxicity, from surveillance to diagnosis to management, useful for both oncologists and cardiologists in their clinical practice. In particular, in the first part of the review, we realize a description of the pathogenetic mechanisms and risk factors of the main cardiovascular irAEs. Then, we focus on the management of ICI-related cardiotoxicity by analyzing five main points: (1) identifying and evaluating the type and severity of the cardiotoxicity; (2) deciding whether to withhold ICI therapy; (3) initiating steroid and immunosuppressive therapy; (4) starting conventional cardiac treatment; and (5) restarting ICI therapy. Finally, we discuss the existing evidence on surveillance for ICI-related cardiotoxicity and propose a surveillance strategy for both short- and long-term cardiotoxicity, according to the most recent guidelines.

## 1. Introduction

Cancer immunotherapies have revolutionized the treatment of both solid and hematologic tumors. Their efficacy relies on the ability to unleash the host immune system against cancer cells [1]. While different costimulatory signals induce T cell activation, this process is, at the same time, counterbalanced by immune checkpoints that prevent an exaggerated immune response [2]. The main lymphocyte immune checkpoints are cytotoxic-T-lymphocyte-associated antigen 4 (CTLA-4) and programmed cell death-1 (PD-1), which bind to B7 and PD-L1, respectively, on antigen-presenting cells. Receptor binding within this co-inhibition pathway down-regulates the immune response by decreasing T cell proliferation and migration or by increasing the number of regulatory T cells. Cancer cells up-regulate expression of these ligands to evade the local immune response [3]. Recently, another evaluated immune checkpoint is lymphocyte-activation gene 3 (LAG-3). LAG-3 is a molecule expressed on the cell surface, in particular on T cells. It causes a downregulation effect on the proliferation of T cells and on the effector function of T cells [4]. Immune checkpoint inhibitors (ICIs) are monoclonal antibodies that block immune checkpoints, thus restoring the immune response against tumor cells.

ICIs have yielded promising results in the treatment of different cancers [5,6]. Their indications have rapidly evolved to include not only late-stage disease but also in the first-line or adjuvant setting. To date, eight agents have been approved by the US Food and Drug Administration for use in a multitude of malignancies: one CTLA-4-blocking antibody (ipilimumab); three PD1-blocking antibodies (nivolumab, pembrolizumab, and cemiplimab); three PD-L1-blocking antibodies (atezolizumab, avelumab, and durvalumab); and one LAG-3–blocking antibody (relatlimab) (https://www.fda.gov/ (accessed on 28 September 2022)).

However, the augmented immune response by ICIs can lead to the development of immune-related adverse events (irAEs) involving several organs, particularly when these drugs are used as combination immunotherapy [7]. The safety profile of ICIs is estimated to range between 54% and 76% [8]. Skin reactions and colitis are the most common irAEs, followed by pneumonitis and hepatitis [1]. Recently, the incidence of IrAEs has increased, in particular following the administration of anti-CTLA4 agents. The IrAEs reported in these patients include skin reactions (44%) and gastrointestinal tract symptoms such as colitis (35%). The irAEs related to the use of anti-PD1 and/or anti-PD-L1 mainly consist of endocrinopathies, such as hypothyroidism (8–10%) and hyperthyroidism (6%) and pneumonitis (2–5%) [9]. These immune-mediated toxicities are largely reversible with the temporary cessation of ICI therapy and can be typically controlled by the administration of glucocorticoids. Conversely, cardiovascular (CV) immune-related complications, while relatively rare, can be associated with significant morbidity and mortality, particularly myocarditis [3,10].

In a recent meta-analysis aimed at estimating the incidence of adverse CV events from immunotherapy, 51 clinical trials and a total of 13,646 patients treated with ICIs in monotherapy, dual therapy, or chemoimmunotherapy were examined. The drugs used included anti-CTLA-4, anti-PD1, and anti-PD-L1 agents. The analysis showed that, in patients treated with an ICI as monotherapy, the overall incidence of CV adverse events was 3.1% (95% CI 0.736–7.06) and, in those treated with dual immunotherapy, 5.8% (95% CI 3.8647–15.5356), thus demonstrating a positive correlation with the number of ICIs. In patients receiving chemoimmunotherapy, the incidence was 3.7% (95% CI 1.6183–9.0977) [11]. In a recent meta-analysis of 32,518 patients receiving ICIs, a higher risk of myocarditis, pericardial illnesses, heart failure (HF), dyslipidemia, myocardial infarction (MI), and cerebral arterial ischemia was reported [12]. A systematic review showed that the most frequent cardiological irAEs were myocarditis (50.8% of all cardiological adverse events) followed by atrial fibrillation and acute HF [11].

In a recent phase 3 study for anti-LAG-3 in combination with anti-PD1 the incidence of myocarditis was 1.7%, and grade 3 or 4 events occurred in 0.6% of patients [13].

To date, with regard to anti-LAG-3 in monotherapy, we do not have sufficient data on cardiotoxicity, but in preclinical studies for this drug on mice, cardiological irAEs have not been described [14]. In the largest case series of patients with ICI-associated myocarditis, the 122 patients had an early onset of symptoms (median of 30 days after initial ICI exposure) [15]. Late CV events (>90 days) are less well characterized but generally carry a higher risk of non-inflammatory HF, progressive atherosclerosis, hypertension, and mortality [12]. The risk of death following the development of immunotherapy-induced myocarditis is between 38% and 46% [16]. Indeed, myocarditis is considered a severe irAE and therefore a cause for permanent discontinuation of immunotherapy according to current guidelines [17,18]. Nonetheless, the occurrence of adverse CV events necessitating the discontinuation of treatment can lead to a worsening of the patient’s prognosis. A re-introduction of immunotherapy after a CV adverse event can be considered only in selected cases (e.g., mild cardiotoxicity, absence of therapeutic alternatives, or response to treatment before the onset of the adverse event) and after discussion within a multidisciplinary clinical team [16,18].

In the following, we provide an overview of the main types of ICI-related cardiotoxicity that clinicians might encounter and summarize current knowledge on the pathogenetic mechanisms and predisposing factors. We also offer a practical model for prevention, surveillance, diagnosis, and management of ICI-related CV events.

## 2. ICI-Related Cardiotoxicity

### 2.1. Pathogenetic Mechanisms

#### 2.1.1. Myocarditis

Although the exact mechanism involved in the development of cardiotoxicity is incompletely understood, evidence suggests a role for the robust proliferation and activation of T cells expressing common, high-frequency receptors against antigens shared by the tumor and affected tissue [19,20]. T cells from the same clones have been found in both the tumor and the inflamed myocardium following immunotherapy. In animal models, CTLA-4, PD1, PD-L1, and LAG-3 protect the heart against immune-mediated injury after stress [4,21,22,23]. ICI inhibition may, therefore, make cardiac cells more vulnerable to damage. Thus, the same T cell response that is therapeutic against the tumor may be responsible for triggering myocarditis (Figure 1).

#### 2.1.2. Pericardial Disease

The same pathogenetic mechanism described for ICI-related myocarditis, i.e., ICI-stimulated cytotoxic T cells, can also give rise to pericardial disease [24] (Figure 1).

#### 2.1.3. Takotsubo-Like Cardiomyopathy

The mechanism underlying ICI-associated Takotsubo cardiomyopathy is unknown but, unlike myocarditis and pericarditis, it seems to be non-inflammatory. ICIs may be directly responsible by causing acute multivessel coronary spasms. Alternatively, their damage may be indirect, the result of adrenergic stress during early ICI therapy in the form of a sudden and massive release of catecholamines from the adrenal glands or postganglionic sympathetic nerves in the heart, causing catecholamine-mediated myocardial stunning [25,26].

#### 2.1.4. Myocardial Infarction

Several pathophysiological mechanisms for ICI-associated MI have been hypothesized, but none have been definitively demonstrated [25]. There is emerging evidence of a link between T cell activity and atherosclerotic plaque stability; because PD1 levels are elevated in plaque T cells, ICI therapy (e.g., PD1 inhibition) might activate these T cells and thus worsen atherosclerotic disease, potentially leading to cardiac ischemic events [27]. Moreover, ICI-associated inflammation may cause the rupture of the plaque’s fibrous cap, resulting in acute MI (Figure 1). Coronary spasm leading to ST elevation following PD1 inhibitor (pembrolizumab) therapy has also been postulated as a mechanism of ICI-associated acute MI [28]. The precise mechanism of coronary spasm is unknown, but it may be linked to a systemic inflammatory response syndrome. Finally, the direct activation of T-cell-mediated coronary vasculitis in the absence of atherosclerosis is a plausible mechanism of ICI-related acute MI, but this has yet to be reported [25].

#### 2.1.5. Arrhythmias and Conduction Disorders

Inflammation is thought to be the primary cause of conduction disease and ventricular arrhythmias, either locally in the ventricle or His-Purkinje system or systemically. Atrial fibrillation can occur as a result of myocarditis, pericarditis, systemic inflammation, or secondary to another irAE, such as thyroiditis [15,25,29] (Figure 1).

### 2.2. Risk Factors

Combination therapy (*p* < 0.001), diabetes (odds ratio [OR] 3.36, *p* = 0.01), obesity (*p* = 0.02), and anti-CTLA-4 therapy (*p* = 0.01) were identified in an international registry as independent risk factors for cardiotoxicity [30,31]. However, the most well-established risk factor for the development of ICI-associated CV IrAEs is combination ICI therapy (OR 1.93, *p* = 0.008) [32,33], which, compared to monotherapy, confers a nearly five-fold higher risk of ICI-associated myocarditis (*p* < 0.001) [19]. Underlying autoimmune disease is potentially a risk factor on its own [34].

Pre-existing CV risk factors and previous CV disease (CVD), such as MI, HF, and previous cancer therapy-induced left ventricular (LV) dysfunction, may also be associated with the development of ICI-associated myocarditis [16,25]. In a multicenter international registry of patients treated with ICIs, those with myocarditis had higher rates of hypertension (60 vs. 48%, *p* = 0.009) and tobacco use (48 vs. 17%, *p* = 0.001) and were more likely to be taking statins (39 vs. 29%, *p* = 0.04) and angiotensin-converting enzyme (ACE) inhibitors/angiotensin receptor blockers (32 vs. 23%, *p* = 0.04) [35]. If T-cell-mediated responses indeed contribute to the progression of acquired heart disease, as previously demonstrated [36], then ICIs could cause the acceleration or decompensation of pre-existing HF in susceptible patients. In addition to these direct effects on myocardial T cell regulation, the general increase in systemic inflammation commonly observed in patients undergoing ICI therapy can contribute to HF progression [25]. However, this scenario remains hypothetical and a clear understanding of the risk factors for ICI cardiotoxicity is still lacking.

According to the latest guidelines of the European Society of Cardiology (ESC) on cardiotoxicity, conditions related to a high ICI-related CV toxicity risk at baseline include: (a) dual ICI therapy (e.g., ipilimumab and nivolumab); (b) combination ICI therapy with other cardiotoxic therapies; and (c) patients with ICI-related non-CV events or prior cancer-therapy-related dysfunction or CVD [18,32,33]. Based on the type and number of predisposing factors, patients should be stratified as at low or high risk and the surveillance protocol should be adjusted accordingly (see below, “ICI-related cardiotoxicity management”). Nonetheless, further studies are required to better identify those patients at highest risk of developing cardiovascular irAEs.

The main potential predisposing factors for ICI-related cardiotoxicity are detailed in Table 1.

## 3. Management of ICI-Related Cardiotoxicity

The management of ICI-related cardiotoxicity consists of: (1) identifying and evaluating the type and severity of the cardiotoxicity; (2) deciding whether to withhold ICI therapy; (3) initiating steroid and immunosuppressive therapy; (4) starting conventional cardiac treatment; and (5) restarting ICI therapy.

### 3.1. Identifying and Evaluating the Type and Severity of the Cardiotoxicity

An integrative clinical approach, beginning with a strong awareness of the patient’s risk, is mandatory to identify and evaluate the different types of ICI-related cardiotoxicity. A complete CV evaluation, including physical examination, blood pressure (BP), natriuretic peptide (NP) (either brain natriuretic peptide [BNP] or N-terminal proBNP [NT-proBNP]), lipid profile, HbA1c, and electrocardiogram (ECG), at baseline and during treatment according to the patient’s risk, is recommended [18]. Multimodality-based cardiac imaging, including standard assessment with ECG and two-dimensional transthoracic echocardiography (TTE), is increasingly being used to better define the cardiotoxicity and may be followed by an advanced evaluation, such as three-dimensional echocardiography and/or myocardial strain imaging and/or cardiac magnetic resonance (CMR). The utility of these methods will depend on the clinical presentation. Indeed, signs and symptoms of cardiac toxicity due to ICIs are variable depending on the type of cardiac involvement and the degree of disease. Cardiotoxicity may present with non-organ specific symptoms such as fatigue, weakness, muscle pain, and syncope. However, typical cardiac symptoms such as palpitations, shortness of breath, chest pain, pulmonary or lower extremity edema, and irregular heartbeat can occur at any time of treatment. Owing to such a wide and heterogeneous presentation, including also asymptomatic states or vague signs and symptoms, the diagnosis of ICI-related cardiotoxicity could also be challenging. The recognition of *subclinical types of cardiotoxicities* might be crucial in this setting. For example, a subclinical rise of cardiac biomarkers in the absence of clear symptoms could be an early sign of cardiotoxicity. In addition, nonspecific symptoms may be obscured by other non-cardiac ICI-related adverse events such as myositis, hypothyroidism, pneumonitis, or other symptoms related to the primary malignancy or coexisting conditions. Furthermore, overall subtle signs and symptoms may become progressive and may need to be adequately interpreted to initiate management and avoid complications. Thus, any relevant alerting symptoms (from a vague malaise to overt symptoms of chest pain, dyspnea, palpitations, pre-syncope, and syncope) or mild ECG/TTE alterations should trigger immediate monitoring with subsequent referral to cardiology/cardio-oncology specialists. For the oncologist, a determination of the severity of the cardiotoxicity is crucial, as it will allow patient-tailored management. Currently, the severity of ICI-associated cardiotoxicity is divided into four grades, as shown in Table 2 [17].

In the following, the incidence, clinical presentation, and diagnosis of the main CV complications are discussed.

#### 3.1.1. Myocarditis

Myocarditis is the most common CV irAE (45%). It is a potentially severe complication with a high fatality (up to 50% of affected patients). An international multicenter registry reported an overall prevalence of 1.14% for ICI-associated myocarditis and 2.4% in patients receiving combination (two or more ICIs) therapy [16,31]. The onset of ICI-related myocarditis is usually during the early phase of ICI treatment (median of 30 days after initial exposure), with the majority of cases occurring within two to three months [31], although late development (after week 20) has also been reported [37]. Cardiotoxicity, however, can occur at any time during ICI therapy and, in some cases, even after the drug has been discontinued [38].

The clinical presentation can range from an asymptomatic elevation of biomarkers to fulminant myocarditis with cardiogenic shock. The most common primary symptom of ICI-related myocarditis is shortness of breath [1]. Further symptoms may include chest pain, fatigue, and myalgias. An international multicenter registry found that nearly half of patients with ICI-associated myocarditis experienced major adverse cardiovascular events (MACE), such as atrial and ventricular arrhythmias, complete heart block, HF, cardiogenic shock, or death [25]. Myocarditis caused by combination ICI therapy is more likely to be severe and has a worse prognosis than myocarditis that develops during ICI monotherapy [19]. As previously stated, the initial work-up for suspected ICI-myocarditis includes cardiac biomarkers, ECG and TTE, followed by CMR and, in selected cases, endomyocardial biopsy (Table 3).

Biomarkers, particularly troponin and BNP, are sensitive diagnostic indicators, but they lack specificity in ICI-associated myocarditis. In a multicenter case-control study, troponin and NT-proBNP levels were elevated in 94% and 66% of patients with ICI-associated myocarditis, respectively [31]. In a descriptive observational analysis based on the medical records of patients from two cardio-oncology units, BNP/NT-pro BNP was increased in 100% of patients diagnosed with ICI-associated cardiotoxicity [39]. Despite this high rate of increased BNP or NT-proBNP levels in ICI-related myocarditis, neither marker is specific since both are also markers of cardiac strain and may thus be elevated in non-inflammatory LV dysfunction. Moreover, BNP and NT-proBNP may be chronically elevated in cancer patients due to cancer-related inflammation [40]. Cardiac troponin (cTn) is likewise frequently elevated in ICI-related myocarditis, but it is not specific, although it does have relevant prognostic implications since higher levels predict a worse prognosis [31].

ECG is usually the first diagnostic evaluation in patients with suspected cardiac damage. Nonetheless, although ECG abnormalities have been observed in 89% of ICI-related myocarditis, in some cases the results may be normal [31]. The main ECG findings range from atrial or ventricular arrhythmias to intraventricular conduction delay up to complete heart block if the inflammatory infiltrates involve the conduction system [19,29]. Sinus tachycardia, T wave inversion, new-onset Q waves, and ST elevation may also be observed.

Echocardiography is the first-line imaging modality used to identify the development of ICI-related myocarditis. While wall motion abnormalities with impairment of LV systolic function are usually observed, in a case-control study less than half of the patients with ICI-related myocarditis had severe LV systolic dysfunction, defined as a left ventricular ejection fraction (LVEF) < 35% [39]. Moreover, in one study, up to 50% of patients with ICI-related myocarditis had a normal LVEF and, among those with MACE, 38%, had normal LV systolic function [31]. Unlike in classical myocarditis, in which a preserved EF is regarded as benign, a normal EF does not rule out ICI-associated myocarditis [24]. Accordingly, in patients with suspected myocarditis who have a normal LVEF on TTE, additional assessments, such as CMR, are recommended to detect any myocardial inflammation. Data are emerging on the role of global longitudinal strain (GLS) in ICI-associated myocarditis. A retrospective study of 101 patients showed that those with ICI-related myocarditis had significantly lower GLS values than determined in controls, with the former including patients with reduced and preserved EF [41]. In that study, a lower GLS value was shown to be strongly associated with the development of MACE, independent of LVEF status [41].

The preferred imaging tool for the diagnosis of ICI-associated myocarditis is CMR. However, as the specific CMR features of ICI-induced myocarditis are not well described, the modified Lake Louise criteria should be applied [25,42]. Early research on CMR in ICI-associated myocarditis revealed its low sensitivity in the detection of myocarditis. In an international registry of 103 patients with ICI-associated myocarditis, late gadolinium enhancement (LGE) was present in 48%, of whom 43% had a preserved LVEF without a specific LGE pattern [43] (Figure 2). In the same study, 50–60% of patients with biopsy-proven myocarditis had negative findings on both T2-STIR and LGE imaging, indicating a relatively low agreement with endomyocardial biopsy and the need for additional tissue characterization parameters, such as T1 and T2 mapping. However, another study demonstrated an increase in the rate of LGE over time, from 21.6% when CMR was performed within 4 days of admission to 72.0% when it was performed later [43]. Thus, the timing of CMR in the detection of ICI-associated myocarditis is likely to be crucial and may account for the limited sensitivity reported in earlier studies. In patients with persistent clinical suspicion but inconclusive CMR findings, or when CMR is contraindicated or not available, cardiac fluorodeoxy–glucose positron emission tomography is an alternative imaging modality that can be used to assess inflammation [3], although its sensitivity is low and it requires a strict 18-h carbohydrate-free fast [44].

Endomyocardial biopsy (EMB) remains the gold standard for the diagnosis of myocarditis [45], but due to its inherent risks and the need for sufficient expertise, it is usually performed only in cases of high clinical suspicion and negative or doubtful non-invasive imaging (e.g., conflicting results of cardiac imaging and biomarkers or in clinically unstable patients) [18]. Typical findings of ICI-related myocarditis include lymphocyte (CD4+, CD8+) and macrophage infiltration and interstitial fibrosis in the affected area. Palaskas and colleagues recently developed a grading system for ICI myocarditis and myocardial inflammation by pathology findings on EMB and noted a correlation with clinical outcomes [46], Table 4. Interestingly, identified patients with EMB confirmed grade 1 ICI induced myocarditis as a low-risk group that may be capable of continuing ICI therapy without immunomodulation. This finding is, however, difficult to routinely introduce to clinical practice given the need for an EMB for grading ICI-related myocarditis. Champion and Stone [47] used EMB to classify ICI-associated myocarditis based on inflammatory cell accumulation in cardiac tissues into high-grade (>50 CD3+ cells/high power field) and low-grade (≤50 CD3+ cells/high power field) groups by EMB finding, Table 4. High-grade patients had a fulminant clinical disease course, while patients with low-grade cell accumulation had a more indolent clinical course. These findings underline the value of EMB assessment of the extent of inflammatory changes in cardiac tissue following ICI, but standardized criteria are yet to be adopted for the histopathologic grading of ICI myocarditis.

#### 3.1.2. Pericardial Disease

Pericardial disease is the second most common CV complication of ICI therapy. In an analysis of adverse drug reactions reported to the World Health Organization’s VigiBase database, the prevalence of ICI-related pericardial disease was 13.6% [48]. In a retrospective study, pericardial effusion developed in 7% of patients with ICI-related cardiotoxicity [38]. Although pericarditis is typically diagnosed within 30 days of ICI treatment initiation [3], in several patients it was discovered several months later [49,50,51,52]. Pericarditis may develop alone or in conjunction with ICI-associated myocarditis (myopericarditis). The reported mortality rate for pericardial disease caused by ICI is 21% [15]. The most common clinical presentation of pericardial disease is shortness of breath, but other symptoms include chest pain, upper venous congestion, and, in patients with cardiac tamponade, cardiogenic shock [48].

The preferred diagnostic tools in patients with clinical suspicion of pericardial involvement are ECG, echocardiogram, and cardiac biomarker determination (Table 3). The clinical diagnosis of pericarditis is usually based on at least two of the following criteria: (i) pericarditic chest pain (typically sharp and improved by sitting up and leaning forward); (ii) pericardial rub; (iii) new ECG changes; and (iv) pericardial effusion [53]. Typical ECG changes include new widespread ST elevation or PR depression. Echocardiography may show pericardial effusion and thus aid in the diagnosis of pericardial tamponade. Additional supporting findings can include elevated markers of inflammation (C-reactive protein, erythrocyte sedimentation rate, and white blood cell count) and evidence of pericardial inflammation by CMR [53]. In the case of concomitant myocarditis, cardiac biomarkers are usually elevated; in these patients, CMR may confirm the diagnosis.

#### 3.1.3. Takotsubo-Like Syndrome

Takotsubo cardiomyopathy has been reported in patients receiving ICI therapy [39,54] and, in one study, it was detected in 14% (4/29) of patients with ICI-related cardiotoxicity [39]. The clinical presentation typically includes acute chest pain, breathlessness, and palpitations due to sinus tachycardia or arrhythmias. In more severe cases, cardiogenic shock may be present.

The diagnostic pathway of Takotsubo syndrome should consist of a clinical examination, ECG, cardiac biomarkers, and echocardiography (Table 3). In most cases, the cTN level is elevated but the rise is disproportionately low relative to the extent of cardiac dysfunction. BNP and NT-proBNP levels are usually elevated as well, but, similar to myocarditis, they lack specificity. The ECG abnormalities in the acute phase of Takotsubo cardiomyopathy can include ST segment elevation, ST segment depression, and new left bundle branch block, with deep and widespread T-wave inversion and significant QT prolongation usually developing 24–48 h later. Echocardiography commonly shows transient and reversible regional wall motion abnormalities, typically involving the apex and medial segments of the left ventricle. Due to its similar clinical and instrumental presentation, acute coronary syndrome should be ruled out by coronary angiography or coronary computed tomography angiography. CMR may help to exclude myocarditis [54].

#### 3.1.4. Acute Myocardial Infarction

Acute coronary syndrome is a rare but possible CV complication of ICI therapy. In a metanalysis of 22 ICI trials in lung cancer, the incidence of ICI-associated MI was 1% [3].

The clinical presentation of ICI-associated MI is that of a classical acute coronary syndrome, usually characterized by persistent chest pain associated with elevated cTn, ECG changes, and regional wall motion abnormalities on TTE. The ECG abnormalities observed in acute coronary syndrome include ST elevation or depression and new T wave inversion. A definite diagnosis is obtained by coronary angiography, which will show severe coronary artery disease [24].

#### 3.1.5. Arrhythmias and Conduction Disorders

Conduction delays, ventricular arrhythmias, and, especially, atrial fibrillation have all been described in the context of ICI therapy [39,55]. These disorders can occur either in conjunction with other irAEs (e.g., ICI-associated thyroiditis with thyrotoxicosis, ICI-associated myocarditis, ICI-associated pericarditis, or ICI-associated severe systemic inflammatory syndromes) or separately. A previous study showed that conduction disorders were isolated in 13% of patients under ICI treatment [39]. Patients who experience palpitations or syncope should undergo a 12-lead ECG or ambulatory ECG monitoring to rule out arrhythmias.

### 3.2. Deciding Whether to Withhold ICI Therapy

The first approach to managing ICI-related cardiotoxicity is withholding ICI therapy. According to the 2021 American Society of Clinical Oncology (ASCO) guidelines, in case of isolated troponin elevation (grade 1), ICI should be withdrawn and the troponin level rechecked 6 h later; if troponin values normalize or the elevation seems to be unrelated to ICI therapy, ICI may be resumed [17]. For cardiotoxicity grades ≥ 2, ICIs should be discontinued [17]. Whether later re-challenge with ICIs is appropriate will depend on the type and degree of the ICI-related cardiac abnormality, as well as the clinical certainty that the cardiac complication is due to ICI therapy as opposed to other causes. While the ASCO guidelines suggest a discontinuation of ICI-therapy for cardiotoxicity above grade 1, the 2020 National Comprehensive Cancer Network guidelines suggest permanently discontinuing immunotherapy only in patients with grade 3 or 4 toxicity [56]. The 2017 consensus statement of the Society for Immunotherapy of Cancer’s Toxicity Management Working Group recommended permanently discontinuing ICIs in patients with grade 4 toxicity but a re-challenge in patients with grade 3 toxicity who have stabilized [57].

According to the ESC guidelines, interruption of ICI treatment is recommended in all cases of suspected ICI-associated myocarditis (any patient developing new cardiac symptoms, new cardiac arrhythmias, new heart blocks, or a new troponin increase following ICI therapy administered in the past 12 weeks) while the appropriate assessments are performed. In patients with suspected but not confirmed myocarditis, once the abnormal findings have resolved a multidisciplinary clinical team should consider the risk/benefit of permanently stopping vs. resuming ICI treatment. Cessation of ICI treatment is recommended for patients with cancer who developed fulminant or non-fulminant ICI-associated confirmed myocarditis [18]. In patients who develop pericarditis, acute myocardial infarction, or Takotsubo-like syndrome, ICI therapy should be interrupted [25].

### 3.3. Initiating Steroid and Immunosuppressive Therapy

Corticosteroid-based immunosuppressive therapy is the initial treatment for irAEs. The 2021 ASCO guidelines recommend managing grade ≥2 toxicities with high-dose corticosteroids (1–2 mg prednisone/kg/day, oral or IV depending on symptoms) [17]. The steroid dose should be tapered slowly over four to six weeks following the normalization of cardiac function or cardiac biomarker levels [58]. In case of no response to high-dose corticosteroids, cardiac transplant rejection doses of corticosteroids (methylprednisolone 1 g every day), and the addition of either mycophenolate, infliximab, or anti-thymocyte globulin, should be considered [17]. In life-threatening cases, abatacept (CTLA-4 agonist) or alemtuzumab (CD52 blockade) may be considered as an additional immunosuppressant [17].

Glucocorticoid treatment failure can be classified as glucocorticoid-refractory and glucocorticoid-resistant. Glucocorticoid-refractory is defined as no improvement or a worsening of related symptoms after the initial use of glucocorticoids, and glucocorticoid-resistant as the recurrence of symptoms induced by a decreased dose of the drug after an initial response [59]. However, the data are scarce and were mostly obtained in different small cohorts of patients with irAEs. According to those reports, the incidence of glucocorticoid-refractory/resistant responses is between 20% and 60% [59,60,61].

Several drugs have been evaluated for use in case of glucocorticoid therapy failure. Alemtuzumab, an antibody against CD52, is currently recommended in the treatment of myocarditis induced by ICI. However, there are few data on its use in this setting. A case report on the use of alemtuzumab in a 71-year-old woman described a rapid resolution of ICI-mediated cardiac immune toxic effects after the failure of other therapies, including pulse methylprednisolone, mycophenolate mofetil, plasmapheresis, and rituximab [62]. However, it should be kept in mind that alemtuzumab can increase the risk of infection, malignancies, and autoimmune disorders. The IL-6 inhibitor tocilizumab may also be used in the treatment of cardiac irAEs. The two published studies reported a normalization of serum inflammatory myocardial biomarkers and the resolution of ICI-induced complete atrioventricular block [63,64]. The biological agent abatacept is a CTLA-4 immunoglobulin fusion protein that binds to CD80/CD86 on antigen-presenting cells, thus inducing T cell anergy [65]. Abatacept induces an effect that mimics the mechanisms of immunotherapy by down-regulating the activation of ICI-related pathways. It therefore represents a viable option for patients with glucocorticoid-refractory ICI-induced myocarditis. In a mouse model of ICI-induced myocarditis, abatacept reduced immune system activation and increased survival [66,67]. Several clinical studies reported the administration of abatacept in ICI-induced myocarditis [68,69,70], with promising results obtained, in particular in the simultaneous treatment of myositis and myocarditis. However, the risk of cancer relapse should also be noted. Two patients in the reported cases had tumor recurrence after abatacept treatment, at three and four months, respectively [69,70]. Infliximab, already used in the treatment of several autoimmune diseases, was shown to also improve ICI-induced myocarditis [19]. Nonetheless, a previous study showed that infliximab treatment after glucocorticoid failure increased the risk of cardiovascular death (50%) compared to the similar use of other immunosuppressive therapies (19%) [71]. Whether inhibitors of tumor necrosis factor (TNF)-α accelerate or enhance ICI-induced heart failure and myocarditis is currently unknown, as is whether the decreased survival rate is related to an excessive inhibition of immune cell activation against tumor growth or to the negative effects caused by anti-TNF-α therapy itself. Regardless, it is clear that glucocorticoid treatment alone is not sufficient in patients with severe life-threatening ICI-induced myocarditis [67]. Further studies on the use of biological agents and other targets in the treatment of myocardial complications ICI-induced are necessary. Other newly developed biological immunosuppressive agents should be further investigated. For example, two clinical trials, currently in phase 2 and 3, are evaluating the use of abatacept in the treatment of ICI-induced myocardial complications. The first results will be published in 5 years *(NCT05335928, NCT05195645).*

#### Management of ICI Related Myocarditis

According to the ESC guidelines on cardio-oncology, to guide the management pathway all cases of ICI-associated myocarditis should be classified according to the severity of the myocarditis: fulminant or non-fulminant, including symptomatic but hemodynamically and electrically stable patients and incidental cases diagnosed at the same time as other irAEs [18].

Patients with cancer and fulminant or non-fulminant ICI-associated myocarditis should be admitted to the hospital and a level 2 or 3 bed and receive continuous ECG monitoring. Treatment of both non-fulminant and fulminant ICI-associated myocarditis with methylprednisolone 500–1000 mg i.v. bolus once daily for the first 3–5 days should be started as soon as possible, after the diagnosis is considered likely, to reduce MACE, including mortality [18]. If clinical improvement is observed (cTn reduced by >50% from peak level within 24–72 h and the resolution of any LV abnormalities, atrioventricular [AV] block, and arrhythmias), oral prednisolone should be substituted, starting at 1 mg/kg up to 80 mg/day. Although the most appropriate weaning-off protocol has yet to be determined, a weekly reduction of oral prednisolone (most commonly by 10 mg per week) under clinical, ECG, and cTn surveillance should be considered. A reassessment of LV function and cTn should be considered when the prednisolone dose is reduced to 20 mg/day, with continued prednisolone weaning by 5 mg per week to 5 mg/day, and a final reduction from 5 mg/day in 1-mg per week steps [18]. Patients with fulminant ICI-associated myocarditis, complicated by hemodynamic and/or electrical instability, require admission to the intensive care unit. Cardiogenic shock should be managed according to current guidelines [18].

A single dose of i.v. methylprednisolone should be considered in clinically unstable cancer patients in whom ICI-induced myocarditis is suspected at presentation but before the diagnosis can be confirmed. Steroid resistant ICI-associated myocarditis is confirmed when the cTn does not drop significantly (>50% reduction from peak) and/or AV block, ventricular arrhythmia, or LV abnormalities persist despite three days of i.v. methylprednisolone plus cardiac treatments. In these patients, second-line immunosuppression should be considered [18]. However, given the lack of data on a specific second-line immunosuppression regimen, the treatment protocol should be decided upon by a multidisciplinary clinical team. Several agents are currently being investigated, including i.v. mycophenolate mofetil, anti-thymocyte globulin (anti-CD3 antibody), i.v. immunoglobulin, plasma exchange, tocilizumab, abatacept (CTLA-4 agonist), alemtuzumab (anti-CD52 antibody), and tofacitinib, with promising results from case series. Caution is advised regarding the use of infliximab in patients with steroid-refractory myocarditis and HF [18].

In patients with pericarditis, immunosuppression with 1 mg prednisone/kg daily can be started [45,48]. In patients with Takotsubo-like syndrome, although there are no general recommendations on immunosuppression, high-dose corticosteroid therapy was shown to be effective in two reported cases [54]. In patients with acute MI, there is no current evidence supporting the use of immunosuppressive therapy in this setting.

### 3.4. Conventional Cardiac Treatment of Cardiac Complications

Cardiac complications should be managed according to the same treatment guidelines used in patients with CVD not related to ICI use [24,25]. Patients with severe HF or arrhythmias should be admitted to a coronary care unit (ESC guidelines).

Regarding the use of conventional cardiac medication in ICI-related myocarditis, systematic data are not yet available. According to the guidelines for the treatment of HF, beta-blockers and ACE inhibitors are indicated in patients with reduced LVEF, but the cardioprotective role of these drugs in preventing or mitigating ICI-related myocarditis has not yet been demonstrated [1,25]. In the absence of contraindications, Lyon et al. suggest using ACE inhibitors in patients with confirmed ICI-related myocarditis and a LVEF of 50% [25].

In patients with pericarditis, colchicine and nonsteroidal anti-inflammatory drugs may be used as additional treatment according to current guidelines on pericardial disease (Table 3) [53]. Pericardiocentesis should be performed in patients with cardiac tamponade.

In patients with Takotsubo-like syndrome, therapy for HF should be started according to guidelines (Table 3) [26,72]. QT-prolonging drugs should be avoided.

In patients with acute MI, treatment should follow the standard of care, including antiplatelet agents, coronary angiography and, if necessary, percutaneous coronary intervention.

Arrhythmias should be treated according to conventional therapy for supraventricular and ventricular tachycardia (beta-blockers, amiodarone, etc.) [73,74,75]. Patients with atrial fibrillation should be placed on anticoagulant therapy and the restoration of sinus rhythm, either pharmacologically or by electrical cardioversion, should be attempted when deemed appropriate. For new conduction delays, pacemaker implantation should be considered [17].

### 3.5. Restarting ICI Therapy

Following appropriate immunosuppression and the resolution of cardiotoxicity in cases of less severe ICI-mediated cardiotoxic effects (e.g., uncomplicated pericarditis or subclinical myocardial dysfunction without conduction disease), it may be appropriate to restart ICI therapy, albeit with close monitoring for recurrence [25]. ICI rechallenge is not recommended in patients with myocarditis, advanced conduction disease, or critical ventricular arrhythmias [1,25]. However, thus far, few studies have examined re-challenge with ICIs after the development of ICI-related cardiotoxicity. A retrospective study suggested that, after the development of irAEs induced by dual therapy with anti-CTLA-4 and anti-PD1 agents, treatment with an anti-PD1 agent alone can be safely reinitiated [76]. A meta-analysis based on a total of 789 ICI rechallenge cases from 18 different cohort studies, 5 case series studies, and 54 case reports evalutated the safety and efficacy of ICI rechallenge after the initial irAE. The results showed a higher risk of developing all grade of irAEs (mild, moderate, and severe) during ICI rechallenge than during initial ICI; but no difference was observed in the incidence of severe irAEs or in in terms of the efficacy of therapy [77]. Patients with mild irAEs may benefit from continuing or restarting ICI therapy, but with intensified cardiac monitoring. The decision should be based on patient status and the severity of the irAE [16].

According to the latest ESC guidelines, a multisdisciplinary team discussion is recommended to review the decision on whether to restart ICI treatment in patients who have recovered from ICI-associated myocarditis and have been weaned from oral steroid therapy. The factors that should be taken into account include the severity of the ICI-associated myocarditis (fulminant vs. non-fulminant vs. asymptomatic), alternative oncology treatment options, treatment of metastatic disease vs. adjuvant/neoadjuvant indications, and a reduction from dual ICI to single ICI treatment if the myocarditis was triggered by the former [18].

In patients with ICI-related MI, following successful treatment, the restarting of ICI therapy after 30 days can be considered in clinically stable patients, but with intensified cardiac monitoring (Table 3) [25,78].

## 4. Surveillance for ICI-Related Cardiotoxicity

No evidence-based protocols for surveillance strategies in patients with ICI-related cardiotoxicity existed until the recent publication of ESC guidelines on cardio-oncology, which include new algorithms for the management and surveillance of patients treated with ICIs. While a baseline CV assessment is strongly advised for all patients, there is no clear evidence supporting the routine monitoring of cardiac biomarkers or routine ECG and echocardiography. However, a baseline evaluation that includes clinical history and risk factors assessment, cardiac biomarkers, and ECG should be performed, especially in patients receiving combined immunotherapies and those with known CVD. A detailed clinical history is important to identify patients at high risk and should query pre-existing CVD, autoimmune disorders, and previous cardiotoxic treatments, such as anthracyclines and antiErbB2 drugs [79,80]. Some centers also perform a baseline echocardiographic exam [25]. In the series reported by Hu et al., surveillance based on troponin levels was performed on a weekly basis for six weeks, since fulminant myocarditis may occur after a median of 30 days after ICI initiation [3]. Lyon et al. measured troponin and BNP levels and performed ECG prior to the administration of ICI cycles 2–4 in their high-risk patients, such as those treated with combination ICI therapy or ICI and another cardiotoxic drug [25]. The authors suggested echocardiography after the second or before the third doses as well as three- and six-monthly echocardiography in patients with abnormal LV function at baseline [25]. Serial echocardiographic screening may also be considered in high-risk patients (combination ICI, ICI in combination with another cardiotoxic treatment, significant pre-existing heart disease) [81,82].

The recent ESC guidelines stated that all patients on ICI therapy should have an ECG and a determination of troponin levels at baseline (class IB), and high-risk patients should additionally undergo TTE at baseline (class IB). However, due to the lack of evidence-based recommendations, the monitoring of ICI therapy is challenging. Once patients are started on ICIs, they should have an ECG, and the cTn level should be checked before cycles 2–4 (class IIaB). If, in both cases, the results are normal, the exams should be repeated every three cycles until therapy completion to detect subclinical ICI-related CV toxicity (class IIaB). A CV assessment including physical examination, BP, lipid profile, and HbA1c level is recommended every three cycles until the end of therapy (class I) [18].

In the JAVELIN trial, which compared avelumab plus axitinib vs. sunitinib, no clinical value was determined for routine TTE monitoring in asymptomatic patients during treatment [83]. However, in high-risk patients and in those with high baseline cTn levels, TTE monitoring may be considered. In patients who develop ECG abnormalities, new biomarker changes, or new cardiac symptoms at any time, a prompt cardio-oncology evaluation is strongly recommended, including TTE for the evaluation of LVEF and GLS, and CMR when myocarditis is suspected

Based on these reports, we propose the surveillance strategy shown in Figure 3. Every patient scheduled for ICI-therapy should undergo a comprehensive cardiac evaluation at baseline. The follow-up exams of these patients should then be performed according to the risk of developing cardiotoxicity, with more frequent cardiac surveillance reserved for patients judged to be at high risk according to the baseline evaluation and for every patient who develops new cardiac symptoms.

## 5. Long-Term Cardiotoxicity

The longer survival of patients who have received immunotherapy has led to the consideration of possible long-term cardiological toxicity (basically atherosclerosis), as known for common chemotherapy drugs. In a multivariate analysis conducted by Drobni et al. that included patients treated with immunotherapy and screened for age, tumor type, and history of major CV events, the onset of atherosclerotic events (MI and ischemic stroke) was assessed in the two years before and after immunotherapy. The authors found an increase in CV events from 1.37 to 6.55 per 100 person-years at two years (adjusted hazard ratio, 4.8 [95% CI, 3.5–6.5]; *p* < 0.001) [84].

The pathogenetic mechanism mediating the increased risk of atherosclerosis in ICI-treated patients is poorly understood but may be related to the chronic inflammatory status that characterizes atherosclerosis and its modulation by ICIs. In particular, ICI therapy includes anti-PD1 drugs and thus targets the PD1-PDL1 pathway, which physiologically down-regulates the pro-atherogenic activity of T cells. Similarly, the overexpression of CTLA-4 in murine cells was shown to correlate with a decrease in the number of T cells with regulatory activity (Treg). Conversely, CTLA-4 inhibition induces an increase in the number of Treg cells in the atherosclerotic plaque [85]. Over time, the pro-atherogenic activity of immunotherapy may be responsible for systemic CV damage. However, considering the relatively recent introduction of ICI therapy, few data are available on its long-term cardiological toxicity. Further studies should include an evaluation of dose-dependent toxicity.

To date, a CV assessment every 6–12 months is recommended in high-risk patient who require long-term (>12 months) ICI therapy (class IC), but this schedule can also be considered in all patients, including low risk patients requiring long-term (> 12 months) ICI therapy (class IIbC) [18].

## 6. Concurrent irAEs

Another open question concerns the association of cardiological adverse events with toxicity in other organs. A study conducted by Moslehi et al. of 101 patients with myocarditis secondary to ICIs showed that 42% had concurrent severe irAEs, of which the most frequent were myositis (25%) and myasthenia gravis (11%) [86].

In a recent case report, an association between myocarditis and hepatitis was described in a patient with triple-negative breast cancer treated with chemoimmunotherapy. After she was started on methylprednisolone therapy, both toxicities declined until complete resolution [87]. Other case reports of concurrent irEAs are available in the literature. Most concern the association of myocarditis with myositis, hepatitis, or pancreatitis [88,89]. The predisposing factors for the concomitant development of multiple irAEs have yet to be investigated but should be determined to aid in the management of patients treated with ICIs.

## 7. The Role of the Cardio-Oncologist in the Molecular Tumor Board

The importance of the close collaboration of the oncologist with a cardiologist specialist in managing cancer patients has become clear, considering the documented cardiotoxicity of many chemotherapy drugs and target therapy agents. The main purpose of a multidisciplinary cardio-oncology team is to ensure that the patient receives the best cancer treatment with the lowest risk of cardiovascular events [90].

A prospective study by Pareek et al. evaluated the outcome of a cardio-oncology service in the United Kingdom for 5 years based on 535 patients. At a median follow-up of 360 days, 93.8% of the patients with LV systolic dysfunction had an improved LVEF (45% pre vs. 53% post; *p* < 0.001) and New York Heart Association (NYHA) class (NYHA III–IV in 22% pre vs. 10% post; *p* = 0.01). All patients with normal LVEF and biochemical or functional myocardial toxicity, and 88% of patients with LV systolic dysfunction, were deemed fit for the continuation of ICI therapy after cardiovascular optimization [91].

The training of a cardio-oncologist, whether his or her previous expertise is in oncology or cardiology, consists of three levels: (i) exposure and basic overview, (ii) the acquisition of advanced clinical experience and knowledge, and (iii) a cardio-oncology fellowship. This could also be the basis for a new medical sub-specialization [92].

## 8. Roadmap to Prevent and Mitigate ICI-Related Cardiotoxicity: Potential Strategies and Ongoing Research Developments

Currently, there are no specific predictors of development of severe or mild cardiac toxicity caused by ICIs. Patients should be informed and alerted on potential development of ICI cardiotoxicity, about signs and symptoms associated with this event, and on the need, especially in high-at risk patients, for an in-depth screening and close surveillance. These procedures represent a crucial component of preventive strategies before starting immunotherapy. Prospective cardiovascular evaluation seems to be important to detect potential cardiotoxicity. However, novel strategies are needed to better identify high-risk patients since conventional risk stratification algorithms, such as the Framingham risk score, may underestimate cardiovascular risk in patients with cancer.

### 8.1. Development and Validating of Prognostic Biomarkers and Cardiac Imaging Findings for Cardiac irAEs

Existing biomarkers for ICI cardiac irAEs have relatively limited clinical data and/or lack extensive validation. Biomarkers that are appropriately sensitive and specific to therapy-induced injury could find applications in subclinical toxicity detection and pre therapy risk stratification for ICI therapy. Moreover, future biomarkers for cardiac irAEs would be specific enough not to arise from the cancer itself. Currently proposed serum biomarkers include high-sensitivity troponin levels (hs-TnI), microRNAs, C-reactive protein, myeloperoxidase, galectin 3, interleukin family molecules including ST2, matrix metalloproteinase, placental growth factor (PlGF), growth differentiation factor 15, peripheral blood mononuclear cell gene expression profile, and human heart-type fatty acid-binding protein [93]. Pre-treatment hs-Tnl levels at a cut-off of 14 ng/L have been demonstrated to predict cardiovascular endpoints and the progression of cardiac involvement in patients receiving Nivolumab [94]. Accordingly, the Stanford Cancer Institute has recently implemented surveillance for ICI-associated myocarditis with hs-TnI assay [95]. Another predictive measure for cardiac irAEs severity following ICI therapy may be the levels of certain microRNAs. Pre-clinical studies have demonstrated an increased frequency and severity of irAEs in murine models deficient in miR-146a, and studies in human subjects have demonstrated an increased risk of severe irAEs in patients on anti PD-1 therapy who have a single nucleotide polymorphism (SNP) in miR-146a [96]. MiR-34a is a critical regulator of myocardial physiology that increases with age and has been associated with cardiac senescence and dysfunction. Through a variety of effects on the NF-kB and KLF4 signaling pathways, miR34a also modulates T cell and macrophage functions such that elevated levels may predispose patients to ICI-related cardiac toxicities [97]. However, further studies are required to confirm the likelihood that these may have utility as prognostic biomarkers for ICI cardiac irAEs. Besides circulating biomarkers, functional and MRI imaging markers have also been proposed to predict ICI toxicities. Cardiac PET scans entail exposure to ionizing radiation, but studies suggest they may be indicated for measuring long-term ICI effects on the heart [98]. Advanced radioscopic imaging techniques may also evaluate myocardial and vascular changes at the molecular level. A recent retrospective study identified septal late gadolinium enhancement as a possible predictor of cardiac event in patients receiving ICIs [99]. Finally, it will be essential to contextualize any findings from circulatory and imaging biomarkers with the specific mechanism of IrAEs. Specific biomarkers and imaging findings, along specific immunological axes, may be candidates as novel “red-flags” of ICI-specific cardiac irAEs.

### 8.2. Utilization of Immune Checkpoint Inhibitors with Reduced Cardiotoxicity

A shift in focus to research and development of novel ICIs which target antigens that are not shared amongst both the myocardium and tumor in question, unlike the current targets PD-1, PD-L1, CTLA-4, and LAG-3, may limit inflammatory reactions against cardiomyocytes. New drugs under investigation include anti-TIM-3 (T cell immunoglobulin and mucin-containing protein 3), anti-VISTA (V-domain Ig suppressor of T cell activation), anti-TIGIT, and anti-BTLA antibodies [100]. These targets have each been shown to restore antitumor immunologic response in preclinical studies, and they are currently under study in humans. The cardiotoxicity of these agents is currently unknown. It is of utmost importance that these ongoing human studies prioritize the assessment of adverse events, including cardiac toxicities in addition to cancer outcomes.

## 9. Conclusions

In this review we aimed to offer to both oncologists and cardiologists a detailed and updated description of the main aspects of ICI-related cardiotoxicity, including surveillance and multidisciplinary management of both clinical and subclinical ICI-related cardiac toxicity.

As a promising approach in several types of cancer, ICI therapy is rapidly expanding in terms of its indications. Nonetheless, although ICI-related cardiotoxicity is relatively rare, it can lead to serious, life-threatening complications. Thus, a close collaboration between the oncologist and cardiologist is of paramount importance to enable the prompt recognition and management of ICI-related CV adverse events. Currently, little is known about the predisposing factors and underlying mechanism of cardiotoxicity secondary to ICI therapy, and there is no clear support for particular strategies to improve the timely detection and management of these complications. In the meantime, the most recent cardio-oncologic guidelines provide the basis for a practical approach to monitor cancer patients on ICI therapy. For a correct approach to the problem, key elements are: (a) stratifying the risk of cardiotoxicity before anticancer therapy is started, (b) closely following high-risk patients, and (c) multidisciplinary team discussions of optimal patient management.

Prospective studies and clinical registries are urgently needed to obtain further insights in this emerging field of cardiotoxicity and thereby improve the outcome and survival of patients receiving ICIs. In this regard, in the multidisciplinary management of immune-related toxicities, the role of the immunologist may also be necessary for the early identification and treatment of irAEs, so near the figures of the oncologist and the cardio-oncologist, the immuno-oncologist should also be included in the next future.

## Figures and Tables

**Figure 1 cancers-14-05403-f001:**
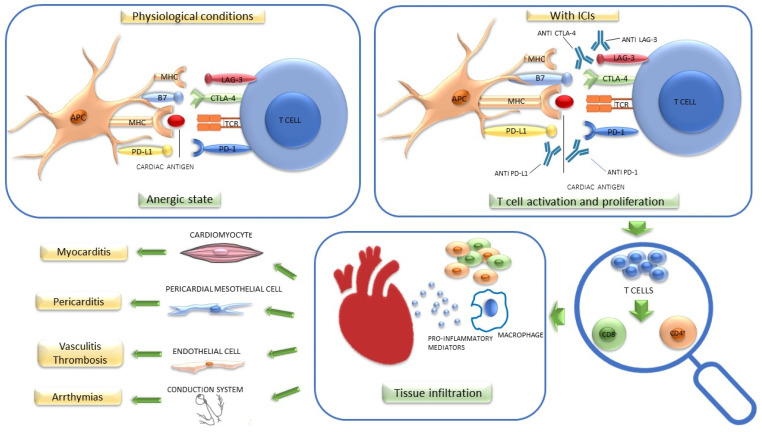
Potential pathogenetic mechanism of cardiotoxicity induced by immune checkpoint inhibitors (ICIs). Under physiological conditions, the activation of the PD1/PD-L1 axis maintains T cells in an anergic state, while CTLA-4 competes with CD80/CD86 to induce a decrease in T cell activation. LAG3 interacts with MHC and upregulates T cells function, downregulates TCR signal transduction, and reduces cytokine production. The inhibition of these immune checkpoints by ICIs may mediate the proliferation of these cells, which, through an antigenic cross-reactivity mechanism, induces cardiac damage on multiple levels. APC: antigen-presenting cell; B7: CD80/CD86; CTLA-4: cytotoxic T-lymphocyte-associated protein; LAG-3: lymphocyte-activation gene 3; MHC: major histocompatibility complex; PD1: programmed cell death protein 1; PD-L1: programmed death ligand 1; TCR: T cell receptor.

**Figure 2 cancers-14-05403-f002:**
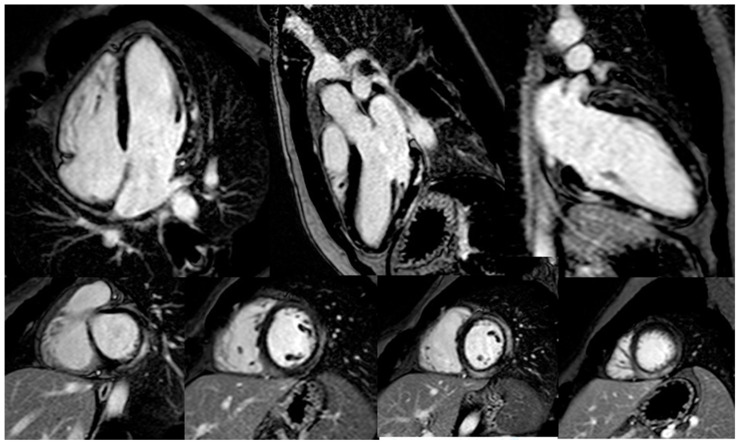
Cardiac magnetic resonance imaging in a patient with ICI-related myocarditis. Late gadolinium enhancement shows a non-ischemic pattern involving several segments of the left ventricle.

**Figure 3 cancers-14-05403-f003:**
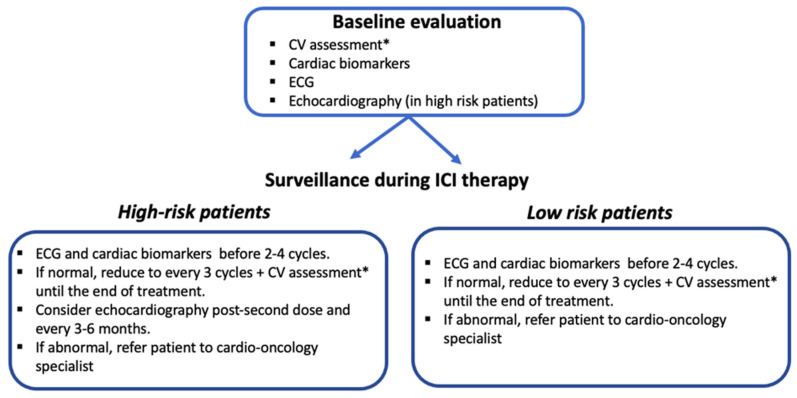
A proposed surveillance strategy in patients undergoing ICI therapy. BNP, brain natriuretic peptide; CV, cardiovascular ECG, electrocardiogram; NT-proBNP, N-terminal pro-brain natriuretic peptide. * consists of a physical examination as well as blood pressure, lipid profile, and HbA1c determinations [18].

**Table 1 cancers-14-05403-t001:** Potential risk factors for cardiovascular adverse events in cancer patients treated with immune checkpoint inhibitors.

Dual immunotherapy Pre-existing cardiovascular risk factors (e.g., obesity, diabetes mellitus, hypertension, smoking) Previous cardiac disease (e.g., myocardial infarction, myocarditis, previous cancer-therapy-related left ventricular dysfunction) Previous anthracycline chemotherapy Concomitant use of other anti-cancer agents (e.g., vascular endothelial growth factor and tyrosine kinase inhibitors Underlying autoimmune disease (e.g., systemic lupus erythematosus, sarcoidosis, rheumatoid arthritis)

**Table 2 cancers-14-05403-t002:** American Society of Clinical Oncology classification of ICI-related cardiovascular toxicities.

Grade	Presentation
1	Asymptomatic, abnormal cardiac biomarker levels, no ECG abnormalities
2	Abnormal cardiac biomarker levels with mild symptoms or new ECG abnormalities without conduction delay
3	Abnormal cardiac biomarker levels with either moderate symptoms or new conduction delay
4	Moderate to severe decompensation requiring IV medication or other intervention, or life-threatening conditions

Adapted and modified from Schneider et al. J Clin Oncol 2021 [17]. ECG, electrocardiography; IV, intravenous.

**Table 3 cancers-14-05403-t003:** Diagnosis and treatment of the main cardiovascular adverse events related to immune check-point inhibitors (ICIs) use.

	Clinical Presentation	Diagnosis	Treatment [18,25]
**Myocarditis**	- Shortness of breath - Chest pain - Pulmonary edema - Cardiogenic shock	- Troponin, NT-proBNP - ECG - Echocardiography - CMR imaging	- Discontinue ICI - Immunosuppressive therapy (Methylprednisolone i.v. 500–1000 mg/day for 3–5 days, then switch to oral prednisone 1 mg/kg/day). If no response: consider second-line immunosuppression - Consider therapy for heart failure
**Pericardial disease**	- Shortness of breath - Chest pain - Cardiogenic shock (in cardiac tamponade)	- ECG - Echocardiography - CMR imaging (to evaluate concomitant myocarditis)	- Withhold ICI therapy - Immunosuppressive therapy (1 mg prednisone/kg/day)) - Consider NSAID and colchicine - Pericardiocentesis if indicated - Consider ICI rechallenge after recovery
**Takotsubo syndrome**	- Chest pain - Shortness of breath - Palpitation - Pulmonary edema - Cardiogenic shock	-Troponin, NT-proBNP - ECG - Echocardiography - CMR imaging - Exclusion of ACS according to ESC and AHA guidelines	- Withhold ICI therapy - No clear evidence on immunosuppressive therapy - Follow management algorithm of Heart Failure Association position statement - Avoid QT-prolonging drugs
**Acute coronary syndrome**	- Chest pain - Shortness of breath - Cardiogenic shock	- Troponin, NT-proBNP - ECG - Echocardiography - Diagnostic algorithm according to ESC and AHA guidelines	- Withhold ICI therapy - No clear evidence on immunosuppressive therapy - Treatment according to ESC and AHA guidelines - Consider ICI therapy rechallenge after > 30 days in stable patients

AHA, American Heart Association; CMR, cardiac magnetic resonance; ECG, electrocardiogram; ESC, European Society of Cardiology; ICI, immune checkpoint inhibitors; NSAID, nonsteroidal anti-inflammatory drug; NT-proBNP, N-terminal pro brain natriuretic peptide.

**Table 4 cancers-14-05403-t004:** Current Pathology Grading criteria for ICI-induced myocardial inflammation.

Grade	Pathological Features
**0**	Negative
**1—Myocardial inflammation**	Multifocal inflammatory infiltrates without overt cardiomyocytes loss by light microscopy
**1A**	Mild inflammatory cell score by immunohistochemistry (10–20 inflammatory cells/high power field)
**1B**	At least moderate inflammatory cell score by immunohistochemistry (>20 inflammatory cells/high power field)
**2—Definite myocarditis**	Multifocal inflammatory cell infiltrates (>40 inflammatory cells/high power field)
*Palaskas et al. Grading Criteria [46]*	
**Grade**	**Immunoistochemistry**
**Low Grade**	50 CD3+ cells/high power field
**High Grade**	>50 CD3+ cells/high power field
*Champion and Stone Grading Criteria [47]*	

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
