# Peer review of "Cardiotoxicity Induced by Immune Checkpoint Inhibitors: What a Cardio-Oncology Team Should Know and Do"

_cancers, 2022, doi:10.3390/cancers14215403_

Round 1

Reviewer 1 Report

In this Review the Authors attract attention to the very important problem of cardiology -  cardiotoxicity of anticancer therapy with immune checkpoint inhibitors including PD-1, PD-L1, and CTLA-4. They provided extensive analysis of the literature describing subclinical, mild and severe criteria of cardiotoxicity.  Moreover, the Authors provided overview of the best current protocols of prevention and treatment.

This very impressive and interesting review can be published.

There are several small concerns:

1. To attract more citations, please, extend Abstract.  It should contain description of all important Chapters of the Review. It would also be good to underline the novelty of the Review in the Abstract and in the Conclusion.

2. If it is possible, could you please pay more attention to the importance of cardiotoxicity prevention and to the description of subclinical manifestations of this chemo cardiac side -effect?

3. Please, read several times the text to catch and improve small typos .. For example, in some laces you introduce abbreviation without unfolding it.  (MI…). Also, it would be good to improve style in some places.

4. Please. Make Fig 1 easier for perception.  It is quite confusing.

Reviewer 2 Report

The authors have summarized the various cardiovascular irAEs induced by ICI exposure --- the latest studies on the pathogenesis, clinical manifestation, diagnosis, and management of ICI-related cardiotoxicity. It’s an important and urgent topic that needs to be fully understood since more and more patients are accepting ICI therapy. A few issues should be addressed to further enhance the quality of the manuscript presented.

1.         The third generation ICI- anti-LAG3 has been approved by FDA early this year. To make the review state of the art, I’d suggest the authors include Lag3 in each section along with anti-PD-1, anti-PD-L1, and anti-CTLA4. Animal studies could be included even though we don’t have abundant evidence in patients on the cardiotoxicity of anti-LAG3.

2.         Could you please expand Table 1 with odds ratio, p-value, No/%, or score?

3.         Cardiac catheterization and heart muscle biopsy are considered the gold standard for the diagnosis of myocarditis through non-invasive imaging, primarily cardiac magnetic resonance, which plays an increasingly important role. This should be included in the manuscript.

4.         Cardiotoxicity induced by ICI is mainly immune-related. It would be nice to include immunology/immunologist/ Cardio-immuno-oncology in this manuscript.

5.         Typos and misspellings: - Page 3, line 117: "MI" instead of "M"?
